# Online Decision Deferral under Budget Constraints

## Abstract

Machine Learning (ML) models are increasingly used to support or substitute decision making. In applications where skilled experts are a limited resource, it is crucial to reduce their burden and automate decisions when the performance of an ML model is at least of equal quality. However, models are often pre-trained and fixed, while tasks arrive sequentially and their distribution may shift. In that case, the respective performance of the decision makers may change, and the deferral algorithm must remain adaptive. We propose a contextual bandit model of this online decision making problem. Our framework includes budget constraints and different types of partial feedback models. Beyond the theoretical guarantees of our algorithm, we propose efficient extensions that achieve remarkable performance on real-world datasets.

## 1 Introduction

Consider a decision-making system that must make investment decisions based on the company's existing resources and on the characteristics of the available assets (the "context"). Although these strategic tasks are usually led by humans, the diversity and complexity of the observable variables have led to the introduction of Machine Learning (ML) models in the loop (Shi et al., 2023). The decision-making system is consequently composite: *human experts* and an *ML model* are working side by side, with possibly nonaligned expertise. The ML model might excel in certain scenarios due to its analytical precision, while human experts bring additional insights and contextual understanding to others. Therefore, it is crucial to design a *deferral meta-model* that can efficiently decide when the decision must be deferred to human experts and when not.

Existing *learning-to-defer* systems, e.g. (Cortes et al., 2016; Madras et al., 2018), often rely on a joint training method where the model and its deferral component are learnt together on large, expert-annotated datasets, with the goal of trading off prediction quality and other criteria such as fairness. When data collection requires highly skilled experts, this traditional approach is very costly. Furthermore, in steadily evolving contexts, such as financial investments or content moderation, the data is subject to constant distribution shifts.

If directly retraining the ML model is not a viable option, then it is desirable to have an adaptive deferral model that estimates and adapts to the current and unknown quality of the ML model and of the human experts.

In this work, we model this scenario as an online two-armed contextual bandit problem that takes into account the budget constraints related to deferring to human experts (Lattimore and Szepesvári, 2020; Agrawal and Devanur, 2014; 2016).

We argue that the resulting online decision deferral system adds significant realism to the problem. We distinguish two types of 'observability' of the decision's performance. In many scenarios, it can be impossible to evaluate alternative decisions post hoc, as an investment or content moderation decision may not allow to observe the counterfactual. This is the case of pure *bandit feedback*, where only the reward of the chosen action (or 'arm') can be observed. Sometimes, however, counterfactual performance can be evaluated: for example, an alternative portfolio performance can be evaluated on market data a posteriori, assuming the investment decision of the agent does not significantly impact these markets. In that case, the observations are richer than the actual performance, and we call it *full information*. We propose to model this problem

with an algorithm that can handle both observability regimes, incorporating additional feedback when it is available.

**Contributions.** We introduce a novel framework for online decision deferral in a budgeted setting with bandit or full information feedback. Our main contribution is to formalize this realistic problem into a tractable online learning setting and thoroughly evaluate the resulting algorithm through experiments on simulated and real data, yielding remarkable results in various settings. Our choice of setting and algorithm allows us to leverage relevant literature and obtain theoretical guarantees on the learnt deferral model.

## 2 Related Work

**Learning to Defer.** Traditionally, deferral was introduced as a rejection mechanism within the model training procedure (Cortes et al., 2016). There, the goal of the system is to minimize the overall cost by rejecting whenever the confidence on the prediction is too low. Learning to defer (Madras et al., 2018) brings the human in the loop by taking into account the accuracy and bias that they could also have on these decisions: a deferral model should balance the uncertainty and biases of any downstream decision maker. This alternative approach has gained attention both among theorists (Madras et al., 2018; Verma and Nalisnick, 2022; Mozannar et al., 2023; Verma et al., 2023) with a focus on uncertainty calibration, and in applications to domains such as medicine (Dvijotham et al., 2022).

Our approach is conceptually similar to the confidence-score method (Raghu et al., 2019; Okati et al., 2021), where a separate confidence score is computed for the human and model predictors. The decision to defer is based on comparing the scores. A weakness of this approach is that the model is not trained concurrently with the deferral system to complement humans where they are most uncertain (Charusaie et al., 2022). However, in the cases where the model is fixed (for practical or legal reasons), it is optimal to defer with a deterministic threshold function (Okati et al., 2021).

**Online learning-to-defer.** The online setting has been noted as an open problem in the learning-to-defer literature (Mozannar et al., 2023). Bordt and Von Luxburg (2022) propose the first online model for learning to *advise*: the machine's decision is always given to the human decision-maker, targeting applications like health care where the decision may only be taken by a human. Joshi et al. (2022) address sequential learning-to-defer in medical settings, proposing a model-based reinforcement learning method to learn the deferral decision; however, they assume offline access to batch data from clinicians. Gao et al. (2021) address the problem of counterfactual risk minimization in human-AI systems.

The question of optimizing decisions online while minimizing resource usage is more common to the online learning and operations research fields: for instance, Cesa-Bianchi et al. (2021) optimize online the Return on Investment, and Jain and Jamieson (2018) learn to manage a crowdfunding platform to maximize the global quality of funded projects. To the best of our knowledge, our model is the first to integrate a human expert in the loop of these online learning settings.

**Bandits with Knapsacks.** Bandits (Lattimore and Szepesvári, 2020) are sequential resource allocation problems with limited information: only the return of the chosen action, or *arm*, can be observed by the learner, and the observed return is often noisy. Knapsack constraints have been extensively studied in the stochastic setting (Madani et al., 2004; Badanidiyuru et al., 2018; Agrawal and Devanur, 2014; 2016; Li and Stoltz, 2022; Slivkins et al., 2023) and rely on online optimization techniques (Hazan and Levy, 2014; Hazan and Li, 2016; Lattimore and György, 2023) to adapt to the cost sequence. There has also been recent interest in the adversarial setting, and proposed algorithms have achieved $O(\log T)$ competitive ratio over the optimal static policy (Immorlica et al., 2022; Sivakumar et al., 2022).

## 3 Formal Problem Setting

We model online learning-to-defer as a two-armed contextual bandit problem with budget constraints. Over a horizon of $T$ interactions, the learner sequentially observes contexts represented by a vector $x_t \in \mathcal{X} \subset \mathbb{R}^d$,

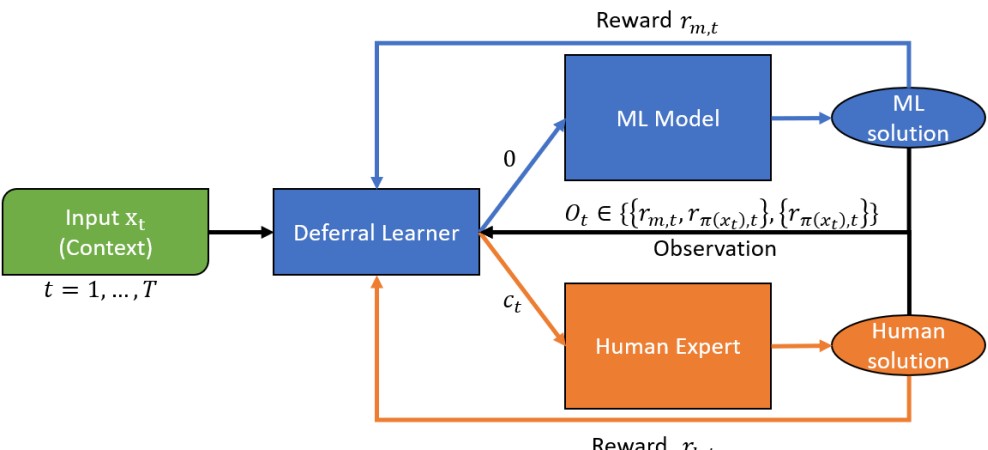

Figure 1: The deferral learner observes inputs online over a time horizon $T$, then decides whether to pay $c_t$ and defer to the human or to let an ML model decide for $c_t = 0$. The deferral learner achieves the reward of the human decision $r_{h,t} = r_h(x_t)$ if the context was deferred to the human, and the ml decision reward $r_{m,t} = r_m(x_t)$ otherwise. The observation $O_t$ is the reward of the decision $r_{\pi(x_t),t}$ in the pure bandit feedback setting. In the full information setting, $O_t = \{r_{m,t}, r_{\pi(x_t),t}\}$.

with $\|x\|_2 \le 1$. They can then decide to assign the task to a human, incurring a cost[1] $c_t \in [0, 1]$ and obtaining the reward $r_{h,t} = r_h(x_t) \in [0, 1]$, or they can predict using the model and obtain the reward $r_{m,t} = r_m(x_t) \in [0, 1]$ at no cost $c_t = 0$. The total budget of the learner is fixed to $B > 0$ and known in advance.

Following the online learning terminology, we denote the (possibly stochastic) policy of the learner as $\pi$ : $\mathcal{X} \to \{m,h\}$. Her objective is to minimize the regret over $T$ rounds compared to a fixed 'good' policy $\pi^*$ whose definition is discussed further below:

$$R_T(\pi) = \sum_{t=1}^{T} r_{\pi^*(x_t)} - \sum_{t=1}^{T} r_{\pi(x_t),t} \quad \text{s.t.} \sum_t c_t \le B \tag{1}$$

Because of the cost constraints, the algorithm must not only estimate whether $r_{h,t} > r_{m,t}$, but also whether the gain $r_{h,t} - r_{m,t}$ justifies the cost expenditure.

As seen in Figure 1, we distinguish two observation models: *Full information* is when the model's performance is always observable but the received reward is that of the chosen decision. The observation is $O_t = \{m, \pi(x_t)\}$ but reward is only $r_{\pi(x_t),t}$; in the *Bandit* setting, the model's performance can only be observed when chosen, *i.e.* $O_t = \{\pi(x_t)\}$.

In the *full information* setting, paying the cost of a human intervention allows to guarantee a human-level performance while the quality of the model can be monitored passively, budget permitting, until we can safely use it when its performance could be superior. In the bandit setting, the model cannot be run in parallel with the human and must be specifically chosen to obtain a performance. In both cases, the cost incurred and the reward obtained are always that of the chosen action only.

**Generalized Linear rewards and costs.** As a first step towards solving this online learning problem, we propose to assume the performance, or *rewards*, of the human and the model follow a generalized linear model. Generalized linear models are an extension of linear regression, where the expected value of the response variable given the predictor variable can be characterized by a link function $\mu$(Li et al., 2017). The two typical examples of generalized linear models are linear ($\mu(x) = x$), and logistic models ($\mu(x) = 1/(1 + e^{-x})$). For each action, there exists an independent and unknown parameter vector $\theta_a^* \in [0, 1]^d$ such that given a context $x \in \mathcal{X}$, $\mathbb{E}[r_a(x)] = \mu(x^\top \theta_a^*)$. The costs and rewards are noisy; at round $t$, with context $x_t$ and decision

---

[1]This cost may be known a priori or not.

$a \in \{m, h\}$, the reward is $r_{a,t} = \mu(x_t^\top \theta_a^*) + \eta_{a,t}$, where $\eta_{a,t}$ is a zero-mean, $\sigma-$sub-Gaussian martingale noise, independent of $x_t$ and $a$. Additionally, the cost of deferring to the human also follows a generalized linear model; $c_t = c_{h,t} = \mu(x_t^\top w^*) + \eta_{c,t}$ for an unknown parameter $w^* \in [0,1]^d$.

Following prior literature(Filippi et al., 2010; Li et al., 2017), we make some regularity assumptions: $\mu$ is twice differentiable, and its first and second derivatives are upper bounded by $L_\mu$ and $M_\mu$. For simplicity, we assume the link function $\mu$ is the same between the reward and cost and between actions. However, the algorithm can extend to varying $\mu$. We also assume there exists a constant $\kappa := \inf_{\|x\| \leq 1, \|\theta - \theta_a^*\| \leq 1} \mu'(x^\top \theta) > 0$. Lastly, we make a regularity assumption on the context distribution: there exists a constant $\sigma_0$ such that $\lambda_{min}(\mathbb{E}_{x \sim D} x x^\top) > \sigma_0 > 0$.

**Defining OPT.** Even when the sequence of rewards and costs are known in advance, finding the optimal solution is a famously NP-hard combinatorial optimization problem (Pisinger and Toth, 1998). When the sequence is not known in advance, the optimal solution is an adaptive policy with full knowledge of the context distribution, the hidden parameters, and its remaining budget. Since the adaptive policy cannot be computed efficiently, the budgeted bandit algorithm is compared against a *static* policy[2] which is only required to fulfill the constraints in expectation(Agrawal and Devanur, 2016).

**Definition 3.1** (Optimal Static Policy). Let $\Pi = \{\pi : \mathcal{X} \to [0,1]\}$ denote the set of policies, where $\pi(x)$ is the probability of deferring to the human on context $x$. Then, define the optimal static policy:

$$\pi^* = \arg\max_{\pi \in \Pi} \mathbb{E}_{x \sim D} \pi(x) \mu(\theta_h^\top x) + (1 - \pi(x)) \mu(\theta_m^\top x)$$

$$\text{subject to } T\mathbb{E}_{x \sim D} \pi(x) \mu(w^\top x) \leq B \tag{2}$$

The optimal value achieved is then $\text{OPT} = T\mathbb{E}_{x \sim D}[\pi^*(x)\mu(\theta_h^\top x) + (1 - \pi^*(x))\mu(\theta_m^\top x)]$

It was shown by Agrawal and Devanur (2016) that the performance of this static policy is an upper bound on the true optimum.

# 4 Algorithm

This problem was studied in the linear setting by Agrawal and Devanur (2016) who consider any number of arms, as well as any number of resources which are expended at different rates. We adapt their algorithm to allow generalized linear rewards and full information/bandit feedback (Algorithm 1). To show that the application of the algorithm to our setting is principled, we present regret bounds as a corollary (Corollary 4.3)

The algorithm is based on the optimism principle and relies on upper confidence bounds on the maximum likelihood estimates of the true but unknown parameters $(\theta_m^*, \theta_h^*, w^*)$ (Filippi et al., 2010; Li et al., 2017; Lattimore and Szepesvári, 2020). This means that, rather than using the greedy estimates, the algorithm uses an *optimistic* one - the parameter which achieves the highest reward (or lowest cost) within the confidence ellipsoid around the least-squares estimate.

At each round, if the budget allows it, we evaluate the potential reward of the model (cost-free) and of the human and its associated cost, and we choose the option with maximal return. Note that we take into account the different observation regimes as described above: the model reward may or not be simultaneously observable to the human's. The set of observed actions at $t$ is denoted $O_t$.

**Definition 4.1** (Maximum Likelihood Estimator). Let $\hat{\theta}_{a,t}$ be the solution to $\sum_{i<t:a \in O_i}(y_{a,i} - \mu(x_i^\top \theta))x_i = 0$. Additionally, let $\hat{w}_{a,t}$ be the solution to: $\sum_{i<t:a \in O_i}(c_{a,i} - \mu(x_i^\top w))x_i = 0$.

**Definition 4.2** (Optimistic estimates). Let $\beta(t) = \frac{\sigma}{\kappa}\sqrt{2d\log\left(\frac{1+2td}{\delta}\right)}$ for a failure probability $\delta \in (0,1)$. Define $M_{a,t} = \sum_{s<t:a \in O_s} x_s x_s^\top$. *Optimistic* estimates of $\theta_a^*$ are computed as follows.

$$\widetilde{\theta}_{a,t} \leftarrow \widehat{\theta}_{a,t} + \beta(t)\frac{(M_{a,t})^{-1}x_t}{\sqrt{x_t^\top (M_{a,t})^{-1}x_t}}$$

---

[2] $\pi$ is independent of the time $t \in [T]$.

---

**Algorithm 1** Generalized Linear Bandit with Budget Constraints (Agrawal and Devanur, 2016)

---

**Require:** Time horizon $T$, budget $B$
**for** $a \in [\text{m, h}]$ **do**
    **Initialize** $M_t^a \leftarrow \mathbf{0}_{d \times d}$, $\widehat{\theta}_{a,t} \leftarrow \mathbf{0}_d, \widehat{w}_{a,t} \leftarrow \mathbf{0}_d$,
**Initialize** $\gamma_t \leftarrow 0.5$, $\alpha \leftarrow 0.5, \epsilon \leftarrow \sqrt{\frac{2}{T}}$
**for** $t = 1, \ldots, \tau = O\left((d + \log 1/\delta)/\sigma_0^2\right)$ **do**
    Play actions at random and update estimates.
**for** $t = \tau + 1, \tau + 2, \ldots, T$ **do**
    Observe context $x_t$, Observation regime $O_t$ (c.f Figure 1)
    **Compute** $\widetilde{\theta}_{a,t}, \widetilde{w}_{a,t}$ per Definition 4.2
    **if** Total Cost Incurred $< B - 1$ **then**
        Play $a_t = \arg\max_{a \in [\text{m, h}]} \mu(x_t^\top \widetilde{\theta}_{a,t}) - \frac{T}{B} \gamma_t \mu(x_t^\top \widetilde{w}_{a,t})$
        Gain reward $r_{a_t,t}$ and incur cost $c_{a_t,t}$
        Observe rewards and costs $\{r_{a,t}\}_{a \in O_t}, \{c_{a,t}\}_{a \in O_t}$ and update estimates.
    Set $g_t \leftarrow \gamma_t(c_{a_t,t} - B/T)$.
    **Define** $\alpha \leftarrow \begin{cases} \alpha(1 + \epsilon)^{g_t} & g_t \geq 0 \\ \alpha(1 - \epsilon)^{-g_t} & g_t < 0 \end{cases}$
    Set $\gamma_{t+1} \leftarrow \frac{\alpha}{1+\alpha}$

---

Similarly, optimistic estimates for the cost function estimates are defined as

$$\widetilde{w}_{a,t} \leftarrow \widehat{w}_{a,t} - \beta(t) \frac{(M_{a,t})^{-1} x_t}{\sqrt{x_t^\top (M_{a,t})^{-1} x_t}}$$

These are optimistic in the sense that for any $x \in \mathcal{X}$, for all $t \in [T]$, $\mathbb{P}(\langle x, \widetilde{\theta}_t^a \rangle \geq \langle x, \theta_*^a \rangle) \geq 1 - \delta$ (see e.g. (Lattimore and Szepesvári, 2020, Chap.20)). Since $\mu$ is an increasing function, this is an upper bound on the expected reward.

The regret guarantees (Corollary 4.3 below) combine the proof of Agrawal and Devanur (2016) with results on unconstrained generalized linear bandits(Filippi et al., 2010; Li et al., 2017). Additionally, we show that the full information setting we propose enjoys the same guarantees. The proof of the regret bounds is deferred to Appendix A.

Note that our setting differs slightly from conventions in prior literature(Agrawal and Devanur, 2016; Filippi et al., 2010; Li et al., 2017). In these works, the algorithm receives arm-dependent contexts $\{x_t\}_a$, and the hidden parameter $\theta^*$ is shared between arms. Though this difference is minor, it means that in our setting there is no shared information between arms: choosing the model does not allow us to gain information on the human's parameter.

**Corollary 4.3** (based on Agrawal and Devanur (2016)). *Under the setting presented in Section 3, and assuming $B > d^{1/2}T^{3/4}$ Algorithm 1 achieves regret $R_T = O\left((\frac{OPT}{B} + 1)\frac{L_\mu d\sigma}{\kappa}\sqrt{T \log \frac{T}{d\delta} \log \frac{T}{d}}\right)$ with probability $1 - \delta$.*

This result means that the method is near optimal since the regret for an unconstrained linear contextual bandit algorithm is always bounded from below by $\Omega(d\sqrt{T})$ (see e.g. (Lattimore and Szepesvári, 2020, Chap. 24)). Note, however, that the budget must be large for this algorithm to perform well. Our experiments are motivated by settings with constant budget (i.e. $B = pT$ for a constant $p \in (0,1]$). However, for some smaller[3] values of $p, T$, *i.e.* for smaller datasets and small budget choices, $pT \leq dT^{3/4}$, and this may affect the convergence of the method.

---

[3]Note that the condition on the budget is in $O()$, indicating that scaling constants are hidden. This discussion refers to orders of magnitudes rather than precise thresholding values.

**Neural Linear Algorithm.** While the assumption of generalized linear costs and rewards is natural for analysis, it may not perform well for every dataset. To address this, we propose a neural variant which applies Algorithm 1 to learned context embeddings. This approach was introduced by Riquelme et al. (2018) in the pure bandit, cost-free setting, where it was shown to have empirical success over the linear algorithm.

We define three separate feed-forward neural networks - one to predict the model reward, one to predict the human reward, and one to predict the cost of deferral. When a context $x_t$ is observed, these neural networks are used to compute three separate embeddings. Each embedding is the output of the penultimate layer of the network, and is passed to deferral algorithm in place of the real context $x_t$. They are used to update the maximum likelihood estimate, context covariance, and optimistic estimate for $\theta_{\rm h}^*, \theta_{\rm m}^*$, and $w^*$ in the main loop of Algorithm 1. Depending on the action chosen, the algorithm may observe $r_{{\rm m},t}, r_{{\rm h},t}$, and/or $c_t$. These are then used to update the neural networks to create more accurate embeddings.

Although this algorithm does not have provable regret guarantees, it is a natural idea that offers a clear comparison to the non-neural generalized linear algorithm. There have been later works that build neural contextual bandits with regret guarantees based on the effective dimension of the neural tangent kernel matrix Zhou et al. (2020); Zhang et al. (2020). We leave the extension of this to the cost-constrained setting to future work.

Note that there are steps that can make this algorithm easier to apply in practice. If compute time is a concern, it is not necessary to train all three neural networks; a viable approach would be to train two embeddings for the human and the model, or even a single embedding for all three parameters. Additionally, we train the networks in batches after a certain number of examples arrive to avoid updating after every step.

## 5 Experiments

The goal of the following experiments is to examine the performance of Algorithm 1 in the expert deferral setting. All experiments take place in the Full Information setting. We defer a comparison to the Pure Bandit setting to Appendix C. Our code and data are available on an anonymous GitHub repository for doubleblind review.

First, to explore how the algorithm learns to exploit the contexts, we examine its performance on artificial data for which the linear model is valid. We compare its performance against a variety of simple baselines.

### 5.1 Synthetic Linear Realizable Data

We tested Algorithm 1 on synthetic data that demonstrate the behavior under different budget constraints and reward functions. We draw the contexts from a discrete distribution with 20 binary features. Let $X = \{x \in \{0,1\}^{20} : \|x\|_1 \leq 8\}$. At each time step, a vector $\bar{x}_t$ is drawn from the the probability distribution $p(x) \propto \lambda^{|x|}$, with $\lambda = 0.3$. Finally, the context is normalized, so $x_t = \frac{\bar{x}_t}{\|\bar{x}_t\|_2}$. This choice of distribution reduces the context space without reducing the dimension, and it mimics applications where the number of nonzero attributes follow a Bell curve.

**Experiment 1: Learning the Optimal Policy** We compare the policy computed by the algorithm against the optimal static policy described in Definition 3.1. The results of this experiment are shown in Figure 2. We see that the regret of Algorithm 1 is sublinear, meaning that the performance of our algorithm gets closer to OPT as $T$ increases.

**Experiment 2: Evolution of Performance with Budget** In the next experiment, we observe how the performance can adapt to different budget constraints. At intermediate budget constraints, the algorithm can explore but must be mindful of limited resources. We demonstrate that Algorithm 1 performs near to optimal regardless of the regime.

The performance in this case also depends on the relative difference between the skills of the human and the model. We investigate two main regimes for which we construct specific parameters $(\theta_m, \theta_h)$. First, the

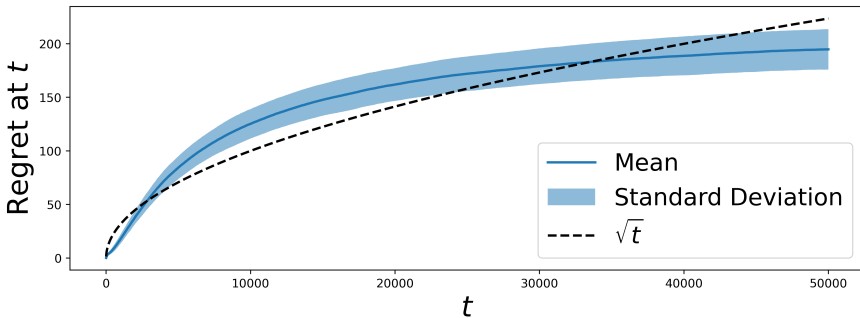

Figure 2: Mean and standard deviation of the regret over 100 trials. The reward and cost functions are sampled uniformly at random from $[0,1]^d$ for each trial.The algorithm is run over $T = 50000$ random contexts with $B = 8000$. Then, the reward received by OPT is computed for the same contexts.

human and the model have the same average reward, but complementary specialties, i.e. $\theta_m = 1 - \theta_h$, with[4] $\theta_h \sim \{x \in \{0,1\}^{20} : \|x\|_1 = 10\}$. Note that this reflects an intuitively ideal setting for our online deferral problem. In the second regime, the human expert has on average a much higher reward than the model, i.e. $\theta_h \sim [0,1]^{20}$ and $\theta_m \sim [0,0.5]^{20}$. In both settings, we compare this algorithm against two simple baselines - ModelOnly, which only uses the response of the model, and ArbitraryHuman, where the algorithm depletes the budget by deferring to the human on arbitrary instances. Additionally, we compare against BestReject, an algorithm that knows the expected model reward on each context and defers to the human if this reward is below a given threshold. We tested thresholds in increments of 0.01, and plotted the best. This algorithm is the optimum among algorithms that do not use any information about the human reward.

The results of this experiment are shown in Figure 3. In both settings, the algorithm significantly outperforms all simple baselines, attaining close to OPT (Definition 3.1).

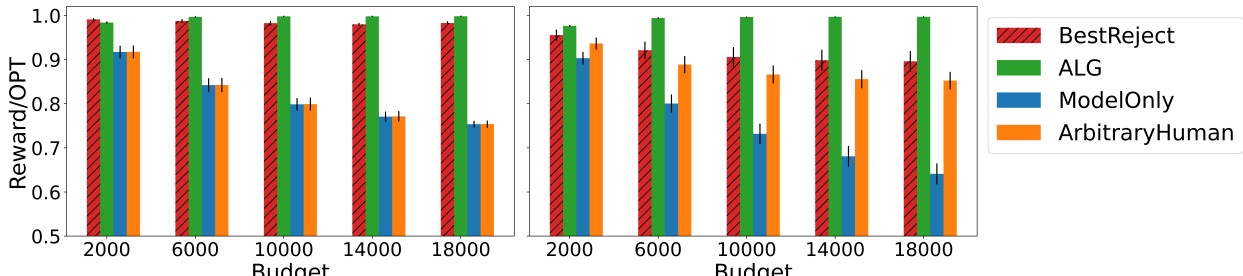

Figure 3: Performance of Algorithm 1 as a percentage of OPT across five different budgets. The two plots correspond to two different human/model reward functions. (Left) $\theta_h \sim \{x \in \{0,1\}^{20} : \|x\|_1 = 10\}$, and $\theta_m = 1 - \theta_h$. (Right) $\theta_h \sim [0,1]^{20}$ and $\theta_m \sim [0,0.5]^{20}$. In both, the cost function is $w_t \sim [0,1]^{20}$. Each experiment runs for $T = 50000$ time steps, and the mean and standard deviation over 20 trials are shown.

In the complementary performance setting, we see that the algorithm does not outperform BestReject by a significant margin. In this setting, the human reward was inversely correlated with the model reward, so the rejection algorithm necessarily deferred to the human on good contexts. In the second setting, the reward functions are uncorrelated, and BestReject has no information on the human reward. It also cannot take advantage of learning the cost function to optimally defer on instances with lower cost.

## 5.2 Real Datasets

In this section, we simulate the online deferral problem on tasks for which performance data was collected from human participants. Unlike with the synthetic data, these experiments are not realizable: human performance data might not admit a good linear approximation. For the following datasets, we use both

---

[4]The symbol $\sim$ denotes 'chosen uniformly at random'.

---
**Algorithm 2** RatioMax4

---
    **Require:**   Capacity $C$, weights $[w_1, w_2, \ldots, w_M]$, values $[v_1, v_2, \ldots, v_M]$
    Set $I = [1, 2, \ldots, M]$.
    Set $w \leftarrow 0$, $v \leftarrow 0$, $c \leftarrow 0$
    **while** $|I| > 0$, $c < 4$ **do**
        Let $i = \operatorname{argmax}_{i \in I} v_i / w_i$
        Set $w \leftarrow w + w_i$, $v \leftarrow v + v_i$, $c \leftarrow c + 1$
        For any $j \in I$ such that $w + w_j > C$, remove $j$ from $I$
    **Return** $v$

---

Algorithm 1 and its neural extension, described in Sec. 4. In this approach, the output of the last hidden layer of a neural network is used as the context $x_t$ when computing the maximum likelihood estimator (Definition 4.1). Further details on our implementation of this algorithm, including the model architectures used, are available in Appendix B.

**Computing OPT.** In real datasets, the context, reward, and cost distributions are not known and cannot be used to compute OPT via Equation 2. Instead, we compute OPT via the following program, where $\pi_t$ is 1 if the optimal policy defers to the human at time $t$, and 0 otherwise.

$$\text{maximize } \sum_{t=1}^{T} \pi_t (r_{\mathrm{m},t} - r_{\mathrm{h},t}) \quad \text{subject to } \sum_{t=1}^{T} \pi_t c_t \leq B \ , \ \pi_j \in [0, 1] \, j = 1, \ldots, 2^d$$

In the infinite budget case, this program simply sets $\pi_t$ to the maximum of $(r_{\mathrm{m},t}, r_{\mathrm{h},t})$. With finite budget, it prefers to defer to the human in cases where the ratio of reward gained $(r_{\mathrm{m},t}, r_{\mathrm{h},t})$ to cost $c_t$ is greater. This can be thought of as the optimal static policy over the empirical distribution defined by the dataset.

**Knapsack Problem Dataset** The 0-1 knapsack problem is a computationally hard combinatorial optimization problem. Given a set of items with specified weights and values, the goal is to choose the set of items with the maximum value whose total weight is less than or equal to the capacity.

In this dataset, 396 participants were asked to solve, via an online form, randomly generated integer instances of the knapsack problem with 18 items. Each participant solved 10 instances, and their performance was recorded as a percentage of the optimal value. The mean performance across all instances was 0.895. They were also given a time limit of 3 minutes, and their time to complete the problem was recorded. This dataset was compiled by Sühr et al. (2024) and we direct the interested reader to their work for more details on data collection, code, study design and the data used for our experiments in this work.

For the model, we used a modified greedy algorithm we call RatioMax4 (Algorithm 2). This algorithm greedily adds the item with the maximum value/weight ratio, until the capacity is reached or four items are added. This algorithm was chosen because it achieves almost identical performance to human participants on average, but it performs well on different instances. This is an ideal setting, since a good online deferral algorithm can achieve large gains by adapting to the relative areas of expertise.

The online deferral problem is as follows. At each time step, an instance of the knapsack problem is presented to the algorithm. The features presented include the weights and values of the knapsack (normalized by capacity), the total capacity, various summary statistics, along with the cumulative average performance and time spent for the current participant. The instances are presented in the same order as they were presented to the participants. The deferral system must decide whether to take the participant's answer (and incur the cost represented by the time they spent on the instance) or to take the answer provided by Algorithm 2. Let $V$ be the value of the answer, and $V_{max}$ be the optimum value for this problem. The reward received is defined to be $\frac{0.1}{1.1 - V/V_{max}}$. This nonlinear reward function incentivizes small improvements near $V/V_{max} \approx 1$.

**Results.** WWe model the reward and cost with a linear link function ($\mu(x) = x$). Figure 4 shows the reward over time for the infinite budget setting. Both algorithms improve significantly over simply predicting with the human or model alone. The regret over time for three different budget constraints (infinite budget,

$B = \frac{1}{2}T$, $B = \frac{1}{4}T$) are plotted in Figure 5. While both algorithms perform similarly in the first few hundred steps, the NeuralLinear algorithm performs better over time.

We observe that the regret appears to grow as $O(t^{2/3})$ over the time scale of the experiment, which indicates that the loss against OPT is sublinear. Note that this is contrary to the theoretical regret of $\tilde{O}(t^{1/2})$. There are several ways in which this experiment differs from the idealized setting. Mainly, the regret bound assumes that the noise in the reward and cost signals are independent of the contexts. This is not generally true in real-world datasets. For example, instances of the knapsack problem that involve more difficult arithmetic may have noisier reward signals. It would be interesting to investigate if a similar trend emerges in other bandit algorithms on real-world datasets.

In the limited budget setting, the regret is substantially higher. With fewer resources, the algorithm cannot explore the human arm as thoroughly, and it has less training data to predict from. In essence, the cost of information is much higher when the budget is limited.

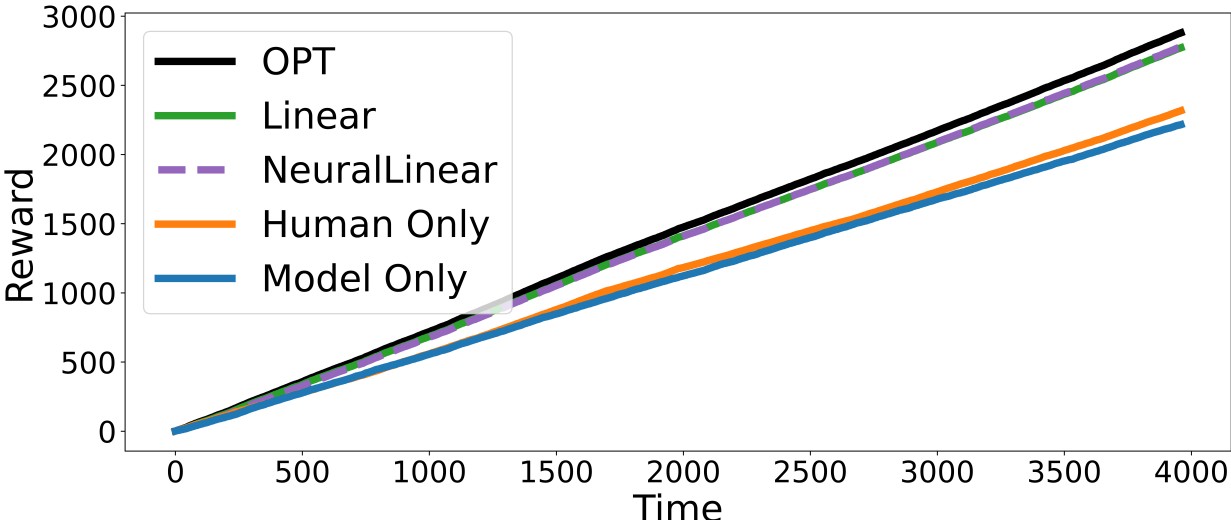

Figure 4: The reward over time of the NeuralLinear and Linear variants on the Knapsack Problem dataset with no budget constraints. The reward of always deferring to the human (HumanOnly) and the model (ModelOnly) are plotted for comparison.

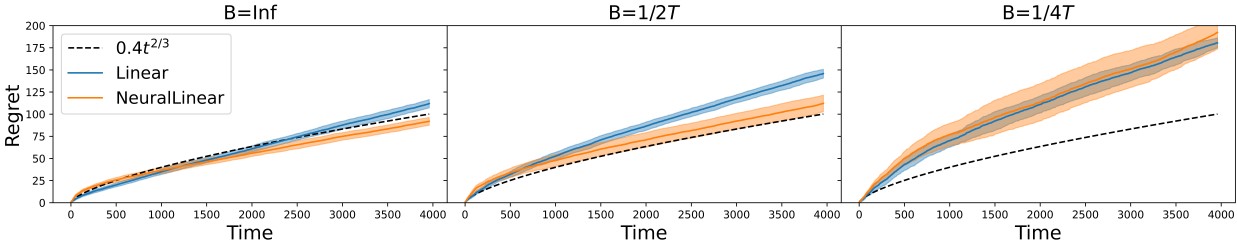

Figure 5: Loss versus OPT of the NeuralLinear and Linear variants of the algorithm on the Knapsack Problem dateset with three different budget constraints: infinite budget (Left), $B = 1/2T$ (Middle), and $B = 1/4T$ (Right). The lines show the mean over 20 trials (rearranging the order of the participants), and the shaded region represents one standard deviation.

**ImageNet 16H Dataset (Steyvers et al., 2022).** This dataset consists of 1200 unique images sampled from a subset of the ImageNet Large Scale Visual Recognition Challenge (Deng et al., 2009), and divided

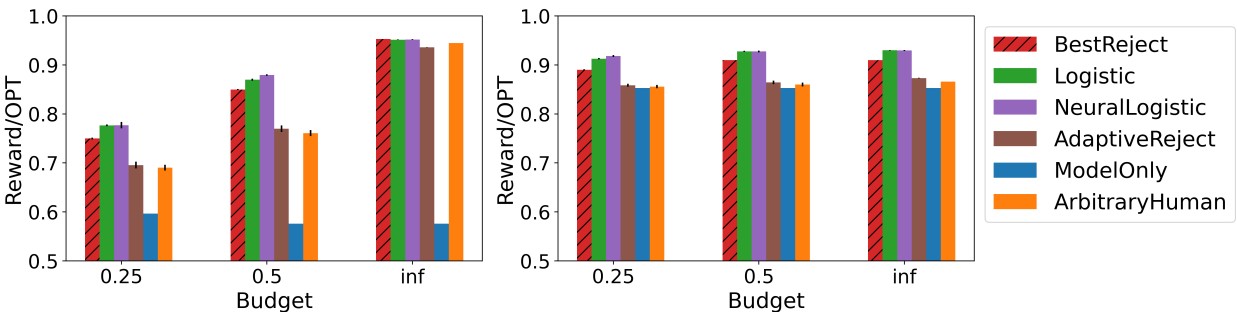

Figure 6: Ratio of total reward to OPT on the ImageNet16H deferral problem, using two different AI models and three different budget constraints. Shows the mean and standard deviation over 20 trials (rearranging the order of the participants). The top plot was generated with the baseline version of Densenet161 (Huang et al., 2017). The bottom plot used the prediction of Densenet161 fine-tuned on noise-augmented images. Note that the BestReject model (striped) uses the best threshold parameter for the dataset and is not possible online.

into 16 categories. The images were perturbed by four different levels of phase noise, resulting in 4800 unique images for classification. These images were then provided to human study participants ($n = 145$), resulting in 28997 total human-classified images. Their confidence and classification time were also recorded. The images were also classified by various commonly used models pre-trained on ImageNet.

We construct the deferral problem as follows. The algorithm receives a reward of 1 if the prediction of the chosen arm was correct, and 0 otherwise. The cost of deferring to the human is the time taken to classify the image (normalized to have mean 1). The features presented to the algorithm include the model classification (encoded as a $16 \times 1$ 1-hot vector), the logits of the model output, and the maximum logit of the output. Since the data from many human participants were combined, we also include the cumulative mean cost and accuracy of the current participant.

**Results.** We model the reward with a logistic link function ($\mu(x) = 1/(1+e^{-x})$), and we model the cost with a linear link function ($\mu(x) = x$). The reward as a ratio of OPT for three different budget constraints are plotted in Figure 6. On this dataset, the NeuralLogistic and Logistic variants of the algorithm achieve similar performance. Since the features already consist of the output of the convolutional model, it is intuitive that the model reward can be predicted using a logistic model. Similarly, the cumulative performance was included as a feature, which could be a simple predictor to differentiate 'good' participants from 'bad' participants. Similarly to the Knapsack data, the algorithm performed significantly worse on the most extreme budget constraint, the baseline model with $B = 0.25T$. On this dataset, the classification times had a long tail, meaning that a large number of participants had a significantly higher cost than the median. The algorithm was not able to properly predict the cost, perhaps indicating that the features were not expressive enough to predict the classification time. This proved to be a significant hindrance in the limited budget setting where the human frequently out-performed the model; deferring to the human as many times as possible was key to achieving performance near OPT.

## 6 Discussions and Conclusions

We introduced a novel online decision making framework that connects learning-to-defer problems with challenges more common to online learning and game theory such as budget constraints, partial observability and unknown performance models. Our algorithmic approach largely leverages existing works in the latter literature but our empirical results show the first concrete results on real data. Crucially, we are able to empirically demonstrate that we can learn how to efficiently assign decisions such that the combined performance is better than either of the predictors alone (model or human).

**Limitations of the model.** One major limitation of the model is the parametric nature of the predictions. While the NeuralLinear model allows us to learn descriptive features, this algorithm does not come with the same theoretical guarantees. It also requires several implementation choices to be made in advance, such as the architecture, learning rate, etc.

We focus on single-resource constraint with single human expert but the original algorithm from Agrawal and Devanur (2016) would naturally extend to allowing for multiple types of resource constraints or multiple experts with partially overlapping specializations.

**Future directions.** Our online approach is designed to readapt a deferral strategy after a possible distribution shift. Going further, our model should be extended to continuously adapt to such changes, for instance by building on recent works on non-stationary bandit models (Liu et al., 2022; Lyu and Cheung, 2023).

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

## A  Proof of Regret Guarantee

For simplicity, we drop the $a$ suffix and look at a single parameter $\theta^*$. Note that all the following also apply to $w$.

Define $C_t = \{\theta : \|\theta - \hat{\theta}_t\|_{M_t} \leq \frac{\sigma}{\kappa}\sqrt{2d\log\left(\frac{1+2td}{\delta}\right)} = \beta(t)\}$. Let $\tau = \min_{t\in[T]} : \lambda_{min}(M_t) \geq 1$. It was shown by Li et al. (2017) that with probability $1-\delta$, $\tau = O\left((d + \log 1/\delta)/\sigma_0^2\right)$ (recalling $\sigma_0 = \lambda_{min}\mathbb{E}_{x\sim\mathcal{D}}xx^\top > 0$).

We present two key lemmas from Li et al. (2017) on generalized linear bandits.

**Lemma A.1** (Lemma 3 of Li et al. (2017))**.** *With probability $1-\delta$, for all $t \geq \tau$, $\theta^* \in C_t$.*

**Lemma A.2** (Lemma 2 of Li et al. (2017))**.** *For all $t > \tau$*

$$\sum_{s=\tau+1}^{t} \|x_s\|_{M_t^{-1}} \leq \sqrt{2(t-\tau)d\log\frac{t}{d}}$$

These three results lead to the following two corollaries, corresponding to two corollaries given by Agrawal and Devanur (2016) in the linear bandit case.

**Corollary A.3** (Corollary 1 of Agrawal and Devanur (2016))**.** *Let $\bar{\theta} \in C_t$. Then,*

$$\sum_{s=\tau}^{T} |x_t^\top\bar{\theta} - x_t^\top\theta^*| \leq \beta(T)\sqrt{2Td\log\frac{T}{d}}$$

*Proof.*

$$\sum_{s=\tau}^{T} |x_t^\top\bar{\theta} - x_t^\top\theta^*| \leq \sum_{t=\tau}^{T} \|\bar{\theta} - \theta^*\|_{V_t}\|x_t\|_{V_t^{-1}}$$

$$\leq \beta(T)\sqrt{2dT\log\frac{T}{d}}$$

The first line comes from a known matrix-norm inequality (Lemma 7 of Agrawal and Devanur (2016)).

The second line comes from Lemmas A.1 and A.2. □

Via the definition of the optimistic estimate:

**Corollary A.4** (Corollary 2 of Agrawal and Devanur (2016))**.** *With probability $1-\delta$, for all $t \geq \tau$, $\mu(x_t^\top\tilde{\theta}_t) \geq \mu(x_t^\top\theta^*)$, and*

$$\sum_{t=1}^{T} \mu(x_t^\top\tilde{\theta}_t) - \mu(x_t^\top\theta^*) \leq L_\mu\beta(T)\sqrt{2dT\log\frac{T}{d}}$$

*Proof.* The first part comes from the assumption that $\mu$ is an increasing function. Thus, $\mu(x_t^\top \tilde{\theta}_t) \geq \mu(x_t^\top \theta^*)$ by Lemma A.1 and the definition of $\tilde{\theta}$ as an optimistic estimator.

The second part follows from the assumption that $\mu$ is $L_\mu$–Lipschitz. So, $\sum_{t=1}^T \mu(x_t^\top \tilde{\theta}_t) - \mu(x_t^\top \theta^*) \leq \sum_{t=1}^T L_\mu(x_t^\top \tilde{\theta}_t - x_t^\top \theta^*)$, and the result follows from Corollary A.3. □

Now we have the tools we need to prove the regret bound.

**Corollary A.5.** *Given $Z$, the algorithm achieves the following with probability $1 - \delta$:*

$$regret(T) = O\left(\left(\frac{OPT}{B} + 1\right)\frac{L_\mu d\sigma}{\kappa}\sqrt{T \log \frac{T}{d\delta} \log \frac{T}{d}}\right)$$

*Proof.* We follow the proof steps presented in Agrawal and Devanur (2016), extending the claims to the generalized linear model when necessary.

Let $T_{stop}$ be the stopping time of the algorithm. Let $R'(T) = O\left(\frac{d\sigma}{\kappa}\sqrt{T \log \frac{T}{d\delta} \log \frac{T}{d}}\right)$. Fix $a$. Also define $T_a = \{\tau < s < T_{stop} : a_t = a\}$. Via the Azuma-Hoeffding inequality,

$$\left|\sum_{s=\tau+1}^{T_{stop}} c_s - \mu(x_s^\top w_{a_t}^*)\right| \leq R'(T)$$

$$\left|\sum_{s=\tau+1}^{T_{stop}} r_s - \mu(x_s^\top \theta_{a_t}^*)\right| \leq R'(T)$$

Additionally, recalling Corollary A.4, with probability $1 - \delta$, $\sum_{t=\tau+1}^{T_{stop}} \mu(x_t^\top \tilde{\theta}_{a_t,t}) - \mu(x_t^\top \theta_{a_t}^*) \leq L_\mu R'(T)$ (and similarly for $w$). Therefore, as in the linear case, a bound on the estimated reward with $\tilde{\theta}$ can serve as a proxy for the bound with $\theta^*$.

Define $\tilde{r}_t = \mu(x_t^\top \tilde{\theta}_{a_t,t})$ and $\tilde{c}_t = \mu(x_t^\top \tilde{w}_{a_t,t})$.

**Lemma A.6** (Lemma 8 of Agrawal and Devanur (2016))**.**

$$\sum_{t=\tau}^{T_{stop}} \mathbb{E}[\tilde{r}_t] \geq \frac{T_{stop}}{T}OPT + Z\sum_{t=\tau}^{T_{stop}} \gamma_t \mathbb{E}[\tilde{c}_t - B/T]$$

*Proof.* Let $a^*$ be the action taken by the optimal static policy at $t$. By Corollary A.4, for any $x_t$, $\mu(x_t^\top \tilde{\theta}_{t,a^*}) \geq \mu(x_t^\top \theta_{a^*}^*)$. Therefore, $\mathbb{E}[\mu(x_t^\top \tilde{\theta}_{t,a^*})] \geq OPT/T$ and $\mathbb{E}[\mu(x_t^\top \tilde{w}_{a^*,T})] \leq B/T$ (taking the expectation over the choice of $x_t$, conditioned on the history). However, since the algorithm chooses the optimal optimistic action:

$$\tilde{r}_t - Z\gamma_t\tilde{c}_t \geq \mu(x_t^\top \tilde{\theta}_{t,a^*}) - Z\gamma_t\mu(x_t^\top \tilde{w}_{t,a^*})$$
$$\mathbb{E}[\tilde{r}_t - Z\gamma_t\tilde{c}_t] \geq \mathbb{E}[\mu(x_t^\top \tilde{\theta}_{t,a^*})] - Z\gamma_t\mathbb{E}[\mu(x_t^\top \tilde{w}_{t,a^*})]$$
$$\geq \frac{OPT}{T} - Z\gamma_t\frac{B}{T}$$

Sum to $T_{stop}$ to get the Lemma statement. □

The rest of the proof follows identically to Agrawal and Devanur (2016). □

# B    Training Details for Neural Algorithm

For both the ImageNet16H data and the Knapsack data, the Neural algorithm trained three separate neural networks with the same architecture. They consist of an input layer with the same dimension as the context, a hidden layer of dimension 50, and a single output layer. Note that this means that in both experiments, the dimension of the linear system is 50. There is a ReLU activation between the input layer and the hidden layer, and a Sigmoid activation on the output. The weights of the networks were updated every 10 steps for the Knapsack data. For the ImageNet16H data, the weights were initially updated every 20 steps, decreasing to every 100 steps after time step 4000. Both were trained with mini-batches of size 500 using the Adam optimizer with learning rate 0.0005 for the Knapsack data and 0.0001 for the ImageNet16H data. The experiments were run on Google Colab servers using their T4 GPU.

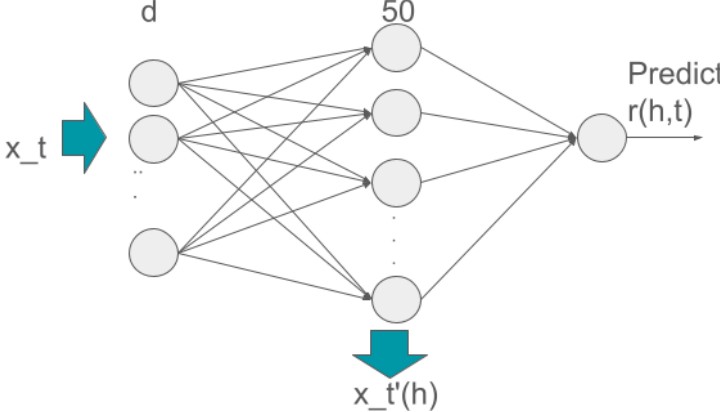

Figure 7: The architecture for computing the embedding for the human arm. An identical architecture exists for the model arm and the cost.

After updating the network weights, the embeddings for all previous contexts are recomputed using the new networks. These new embeddings are used to recompute the estimated parameters per Definition 4.2 and 4.1. As noted in (Riquelme et al., 2018), this may not be practical in applications where all previous contexts cannot be stored, either due to space constraints or legal concerns. In these settings, one can continue to use the previous embeddings and apply weights which decrease the influence of old embeddings on the linear system over time.

# C    Bandit Feedback Experiments

Overall, we do not observe a significant difference in performance between the bandit feedback setting and the full information setting. With random reward and cost functions, the average performance in the full information setting is slightly better, as seen in Figure 8. Interestingly, in the Knapsack dataset, the linear algorithm seemed to perform slightly better in the pure bandit setting (as shown in Figure 9). This may indicate that the full information setting overexplored the human arm.

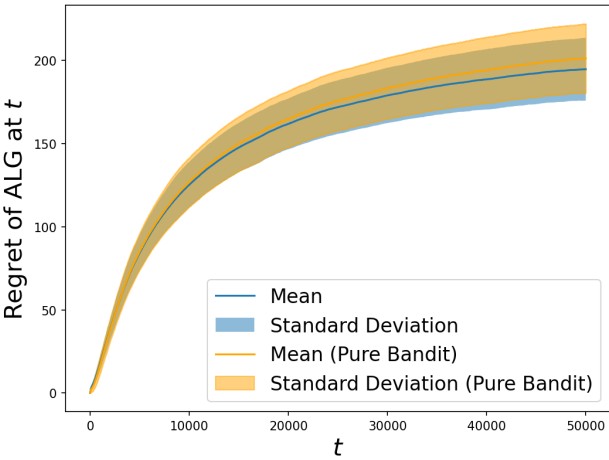

Figure 8: The experiment described in Figure 2 with the Pure Bandit setting included. Mean and standard deviation of the regret over 100 trials. The reward and cost functions are sampled uniformly at random from $[0, 1]^d$ for each trial. The algorithm is run over $T = 50000$ random contexts with $B = 8000$. Then, the reward received by OPT is computed for the same contexts.

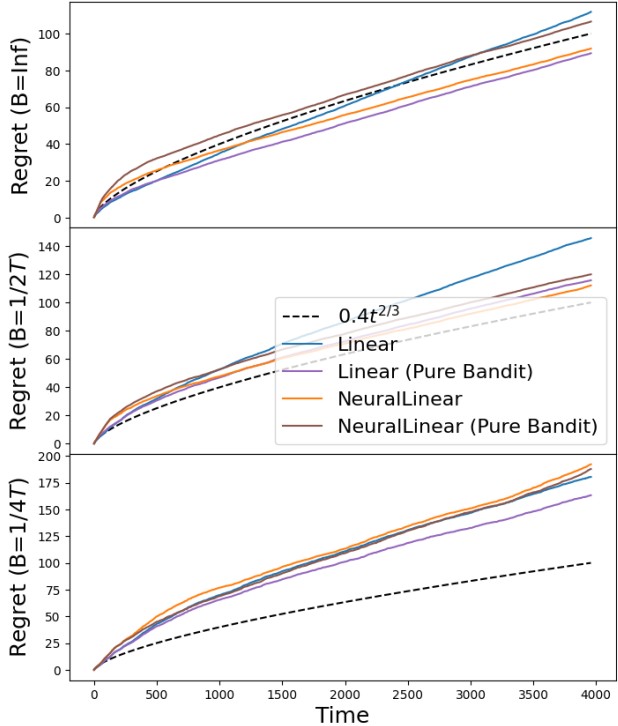

Figure 9: The experiment described in Figure 5 with the Pure Bandit setting included. For clarity, only the means are plotted.

