# OpenReview forum: "Online Decision Deferral under Budget Constraints"
_TMLR — Rejected by TMLR_

### Review · Reviewer_xkep · 2025-03-11

**Summary Of Contributions:**

This paper considers an online learning problem where the deferral agent needs to decide whether to adopt a ML model or a human expert to obtain the reward, given the contextual information. It formulates this problem into a contextual bandits problem under a budget constraint. An bandit algorithm for the generalized linear bandits under budget constraint is proposed with theoretical guarantees. It is further extended to the neural bandits case in order to cope with more complicated scenarios. Empirical experiments are conducted on synthetic and real-world datasets.

**Audience:**

Yes

**Broader Impact Concerns:**

None.

**Claims And Evidence:**

Yes

**Requested Changes:**

It would be beneficial if the authors could provide additional theoretical results, such as guarantees for neural linear bandits, which would be of interest to more researchers.

Unlike emerging fields such as large language models (LLMs), bandit algorithms are well-established, and both theoretical guarantees and empirical performance are essential for evaluating the validity of a proposed approach. While I acknowledge that a gap may exist between theory and practice for many bandit algorithms, this gap is typically small. As such, a paper that places heavy emphasis on empirical results alone may appeal to a limited audience within the bandits research community.

**Strengths And Weaknesses:**

**Strengths**
- The problem formulation is motivated by real-world applications.
- The experiments on several synthetic and real-world datasets show that the proposed algorithm perform well in practice. Thus, this paper might be of interest to researcher focused on the empirical aspects of the online learning algorithms.

**Weaknesses**
- The theoretical contribution is limited, with Corollary 4.3 being the only theoretical result. This corollary extends the regret bounds for linear contextual bandits (Agrawal and Devanur, 2016) to generalized linear bandits using techniques from Li et al. (2017). However, since this extension follows a straightforward application of existing methods, the technical contributions may not be interesting to the researchers focused on theoretical aspects.
- It would be valuable if the authors could provide regret guarantees for the neural linear algorithm, as this would likely be of interest to the broader online learning research community.

---

### Review · Reviewer_wkrD · 2025-03-11

**Summary Of Contributions:**

The paper formulates the online decision deferral in a budgeted setting with bandit or full-information feedback. It conducts many evaluations of the proposed algorithm, on several datasets.

**Audience:**

Yes

**Broader Impact Concerns:**

The paper studies the problem of decision making with budgets. The proposed method might be used to make investment decisions. I don’t have any ethical concerns.

**Claims And Evidence:**

Yes

**Requested Changes:**

1. Page 4: “Defining OPT”, what is OPT short for?

2. In Definition 3.1, since the action set is discrete, i.e.,  $\lbrace m,h \rbrace$, then the policy class $\Pi = \lbrace\pi: X \rightarrow [0,1] \rbrace$ does not make sense. There should be some typos. The writing could be improved. For example, “the performance of this static policy is an upper bound on the true optimum.” What is performance, and optimum? I think the definition is very ambiguous.

3. Since Algorithm 1 works for generalized linear bandit, which is different from that in  (Agrawal and Devanur, 2016). It seems improper to directly refer to that work in the algorithm name.

4. On page 6, there is the neural linear algorithm part. I feel that the message the authors hope to convey is very ambiguous. Maybe some graphs, equations or a detailed explanation can help. Moreover, if there is no theoretical guarantee, it may be better to reorganize the structure and put it in the experiment part.

**Strengths And Weaknesses:**

Strengths：

(1)The paper applies an algorithm with good theoretical guarantees to applications.

(2)The experiments are comprehensive, including both synthetic data and real data.


Weaknesses:

(1) The Algorithm part of this paper is very direct, if it shown to an expert to bandit theory. As is remarked by the authors, the result is a simple combination of Agrawal and Devanur (2016) with results on unconstrained generalized linear bandit.

(2) Since the problem is motivated by the online learning-to-defer, the authors only consider two actions $\{m,h\}$. However, in the field of bandit theory, this assumption is very restrictive. There are works that can deal with similar problems with a larger action set. As a result, I find it difficult to situate this work within the broader bandit theory literature and to identify its key contributions.

(3) As far as I know, it misses some recent related works, for example, [1][2]. There should be a comprehensive comparison with the literature.

(4) Some parts of the writing are not very rigorous. I place some examples in the following section.

[1]  Bandits with Knapsacks and Predictions, Drago et al. 2024.

[2]  On Stochastic Contextual Bandits with Knapsacks in Small Budget Regime, Guo et al. 2025

---

### Review · Reviewer_dSo7 · 2025-03-22

**Summary Of Contributions:**

This paper addresses a type of budget-constrained two-armed contextual bandit problem. In this setting, the two actions correspond to having a pre-trained machine learning model perform a task or having a human perform it, with the latter incurring a cost. The models for both the cost and the reward obtained from performing the task are assumed to follow generalized linear models but are initially unknown. We propose an algorithm for this problem and provide an upper bound on its regret. Furthermore, we validate the effectiveness of the proposed algorithm through numerical experiments.

**Audience:**

Yes

**Claims And Evidence:**

No

**Requested Changes:**

In my view, addressing the concerns regarding correctness outlined above is essential for the paper to be considered for acceptance. However, given the number and severity of the issues, I believe they cannot be resolved through minor revisions.

**Strengths And Weaknesses:**

This paper raises several concerns from the perspective of theoretical soundness. First, Definition 4.1 defines the "maximum likelihood estimator" as the solution to the equation $\sum_i (y_{a,i} - \mu(x_i^{\top} \theta))x_i = 0$. However, this expression corresponds to the MLE only in specific settings, such as when the observations follow an exponential family distribution. In the context of this paper, the validity of this estimator as an MLE is not guaranteed. As described in Section 3, “Generalized Linear Rewards and Costs,” the paper considers generalized linear models—meaning $\mu$ may be nonlinear—and assumes only that the noise is zero-mean $\sigma$-Gaussian. Under such assumptions, the estimator defined in Definition 4.1 is generally not the MLE, nor is it clear whether it is consistent. (In particular, when $\mu$ is nonlinear and the noise is Gaussian, the true MLE takes a different form from that in Definition 4.1.) Furthermore, the paper contains no mention of the exponential family, which would justify the stated form.

Similarly, the "optimistic estimates" defined in Definition 4.2 are claimed to satisfy the following property: "for any $x \in \mathcal{X}$ and for all $t \in [T]$, $P(\langle x, \tilde{\theta}\_{\*}^a \rangle \ge \langle x, \theta^{a}\_{\*} \rangle) \ge 1 - \delta$ (see, e.g., Lattimore and Szepesvári, 2020, Chapter 20)." However, this claim is questionable. Chapter 20 of the cited book discusses confidence bounds for least squares estimators in linear models; it does not address generalized linear models. Even if the discussion were restricted to linear models, the probability bound assessed in the book differs from the one evaluated in this paper, and it is unclear how the cited result justifies the claim made.

Moreover, the proofs included in the Appendix do not appear to rigorously establish the validity of the algorithm presented in the main text. While the authors reference results such as Corollary 2 from Agrawal and Devanur (2016), that work deals specifically with linear models and cannot be directly applied to the analysis of generalized linear models. Notably, the statement labeled “Corollary A.4 (Corollary 2 of Agrawal and Devanur (2016))” on page 13 of the paper differs from the actual Corollary 2 in Agrawal and Devanur (2016).

In light of the above points, I have significant concerns regarding the theoretical soundness of this paper, and therefore cannot support its acceptance under the TMLR acceptance criteria. That said, the problem setting is natural, and the application is clearly described, suggesting that the paper may still be of interest to some readers.

---

> ### Author Response · Authors · 2025-04-04
> **Misunderstanding of theoretical framework**
>
> Thank you for your response. There seems to be a misunderstanding of our definitions.
> With regards to the first comment, we operate in the same generalized linear setting as Li et al "Provably optimal algorithms for generalized linear contextual bandits". We refer you to this paper for the use of the maximum likelihood estimator.
>
> For the comment on the optimistic estimates, we state "These are optimistic in the sense that for any $x \in \mathcal{X}$, for all $t\in [T]$, $P(\langle x,\tilde{\theta}_t^a\rangle \geq \left\langle x, \theta_\ast^a  \right \rangle )\geq 1-\delta$ citing Lattimore.
>
> This is the upper confidence bound on $\theta^*$, and the idea is not dependent on the derivation of $\hat{\theta}$ via generalized linear or linear estimates. We refer to this idea informally in the definition, and refer to the idea more formally in Lemma A.3, referring to Lemma 3 of Li et al.
>
> In the appendix, when we write “Corollary A.4 (Corollary 2 of Agrawal and Devanur (2016))”, we mean that this is the generalized version of the corollary from the 2016 paper. We fully prove the statement in the generalized setting in the appendix. We acknowledge that this is confusing, and will modify the statement to say that the proof merely proceeds along the same lines as Corollary 2 of Agrawal and Devanur .

---

> > ### Comment · Reviewer_dSo7 · 2025-04-18
> >
> > > Thank you for your response. There seems to be a misunderstanding of our definitions. With regards to the first comment, we operate in the same generalized linear setting as Li et al "Provably optimal algorithms for generalized linear contextual bandits". We refer you to this paper for the use of the maximum likelihood estimator.
> >
> > Thank you for your response. First, could you clarify how you are interpreting the definition of the maximum likelihood estimator (MLE)? Let me restate my point.
> >
> > The standard definition of the maximum likelihood estimator is that it is the parameter that maximizes the likelihood function. This is the conventional understanding found in most standard textbooks on mathematical statistics (see, for example, Definition 6.1.1 in [1]).
> >
> > In contrast, the estimator given in Definition 4.1 of your paper does not necessarily maximize the likelihood function. For it to qualify as a maximum likelihood estimator, certain conditions must be met. For instance, the reward distributions would need to belong to an exponential family. In fact, as stated at the beginning of page 4, the model assumes that an arbitrary sub-Gaussian martingale noise independent of $x_t$ and $a$ is added to the rewards. This implies that the reward distribution does not necessarily belong to an exponential family. On the other hand, the papers you cite as [2] (Li et al.) and [3] explicitly mention that their reward distributions lie in an exponential family, and thus the estimators used in those works are valid MLEs under that assumption.
> >
> > However, in your paper, the assumption on the reward distribution has been relaxed and is not limited to exponential families. As a result, the estimator given in Definition 4.1 is not necessarily a maximum likelihood estimator in general. For this reason, I believe it may be inappropriate to label it as such in the definition.
> >
> > Moreover, given this potential mismatch, I am concerned about whether it is valid to apply lemmas and results from prior work—such as those in [2] and [3]—without additional justification under your broader setting.
> >
> >
> > > For the comment on the optimistic estimates, we state "These are optimistic in the sense that
> > > for any $x \in \mathcal{X}$, for all $t \in [T]$, $P(\langle x,\tilde{\theta}\_t^a\rangle \geq \langle x, \theta\_\ast^a  \rangle )\geq 1-\delta$
> > citing Lattimore.
> >
> > > This is the upper confidence bound on $\theta^*$, and the idea is not dependent on the derivation of $\hat{\theta}$ via generalized linear or linear estimates. We refer to this idea informally in the definition, and refer to the idea more formally in Lemma A.3, referring to Lemma 3 of Li et al.
> >
> >
> > It still remains unclear to me how this claim ($\star$): [for any $x \in \mathcal{X}$, for all $t \in [T]$, $P(\langle x,\tilde{\theta}\_t^a\rangle \geq \langle x, \theta\_\ast^a  \rangle )\geq 1-\delta$] is actually established. While I see that you are claiming ($\star$) follows from Lemma A.3, it would be helpful if you could explicitly clarify how exactly this is derived from Lemma A.3.
> >
> > As you define
> > $$
> > \tilde{\theta}\_{a,t} = \hat{\theta}\_{a,t} + \beta(t) \frac{(M\_{a,t})^{-1}x\_t}{\sqrt{ x\_t^\top (M\_{a,t})^{-1}x\_t }}
> > $$
> > in Definition 4.2,
> > I assume that $\beta(t) \frac{x^\top (M_{a,t})^{-1}x_t}{\sqrt{ x_t^\top (M_{a,t})^{-1}x_t }}$ corresponds to the bonus term for $\langle x, \hat{\theta}_t^a \rangle$. However, given that this is not necessarily nonnegative (when $x \neq x_t$) in general, this claim ($\star$) appears quite questionable to me...
> >
> > Moreover, if the result is in fact based on that of Li et al., it would be more appropriate to cite Li et al. instead of Lattimore’s book here. (I cannot find upper confidence bounds for generalized linear models in this book.)
> >
> >
> >
> > * [1] Hogg, R. V., McKean, J. W., & Craig, A. T. (2019). Introduction to mathematical statistics. 8th edition.
> > * [2] L. Li, Y. Lu, and D. Zhou. Provably optimal algorithms for generalized linear contextual bandits. In International Conference on Machine Learning, pages 2071–2080. PMLR, 2017.
> > * [3] S. Filippi, O. Cappe, A. Garivier, and C. Szepesvári. Parametric bandits: The generalized linear case.  Advances in neural information processing systems, 23, 2010.

---

### Author Response · Authors · 2025-04-04
**Response to Reviews**

Thank you to all reviewers for the thorough and constructive responses.

The contribution of our paper lies in proposing a new model for online Human-AI collaboration. Previous models for this problem often assume knowledge about the human expert beforehand, which can be cumbersome in limited data and cost-constrained settings. We propose to leverage existing techniques in the Bandit literature to solve this emerging problem. In addition, we make a small theoretical contribution in generalizing the cost-constrained linear bandit algorithms, but this is not the main contribution of the paper.

---

### Decision · Action_Editor_JsCD · 2025-04-30

**Recommendation:** Reject

**Comment:**

The paper studies a budget-constrained two-armed contextual bandit problem, where the arms correspond to whether to defer decision making of an ML model to a human expert. The work proposes a reasonable algorithm and shows it works well in comprehensive experiments on both synthetic and real data. While the paper has good potentials, reviewers had several major concerns. First, a reviewer raises technical concerns that were not addressed satisfactorily in discussion. Second, given extensive studies of related algorithms, the algorithm and its empirical performance are expected. Given these, all reviewers agreed that the paper is not ready for publication. But the authors are encouraged to strength the paper for resubmission.

**Audience:**

In the current form, the group of interested individuals might be limited.

**Claims And Evidence:**

There are some (fixable) gaps in the theoretical justification. The empirical justification is solid.

**Resubmission Of Major Revision:**

The authors may consider submitting a major revision at a later time.